# GeoCAD: Local Geometry-Controllable CAD Generation with Large Language Models

**Zhanwei Zhang**[1,*] **Kaiyuan Liu**[1] **Junjie Liu**[2] **Wenxiao Wang**[4] **Binbin Lin**[4,†]
**Liang Xie**[3] **Chen Shen**[2] **Deng Cai**[1]
[1] State Key Lab of CAD&CG, Zhejiang University
[2] Alibaba Cloud Computing, [3] Zhejiang University of Technology
[4] School of Software Technology, Zhejiang University
zhanweizhang@zju.edu.cn

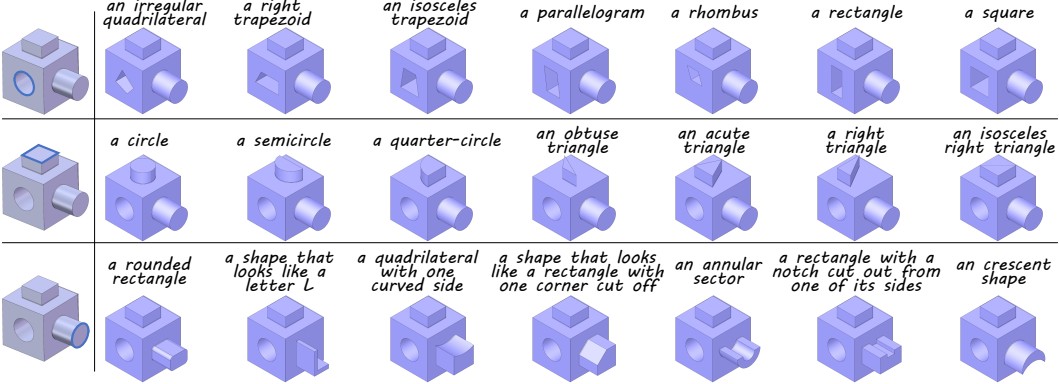

Figure 1: Local geometry-controllable CAD generation achieved by GeoCAD. The input comprises: (1) an original CAD model (the left side), (2) the local part to be modified (highlighted in blue), and (3) user-specific geometric instructions. Subsequently, GeoCAD outputs the revised CAD models where only the target part is altered while adhering to the provided geometric instructions.

## Abstract

Local geometry-controllable computer-aided design (CAD) generation aims to modify local parts of CAD models automatically, enhancing design efficiency. It also ensures that the shapes of newly generated local parts follow user-specific geometric instructions (*e.g.*, an isosceles right triangle or a rectangle with one corner cut off). However, existing methods encounter challenges in achieving this goal. Specifically, they either lack the ability to follow textual instructions or are unable to focus on the local parts. To address this limitation, we introduce GeoCAD, a user-friendly and local geometry-controllable CAD generation method. Specifically, we first propose a complementary captioning strategy to generate geometric instructions for local parts. This strategy involves vertex-based and VLLM-based captioning for systematically annotating simple and complex parts, respectively. In this way, we caption ∼221k different local parts in total. In the training stage, given a CAD model, we randomly mask a local part. Then, using its geometric instruction and the remaining parts as input, we prompt large language models (LLMs) to predict the masked part. During inference, users can specify any local part for modification while adhering to a variety of predefined geometric

* Internship work at Hangzhou YunQi Academy of Engineering and Alibaba Cloud Computing.
† Corresponding author

39th Conference on Neural Information Processing Systems (NeurIPS 2025).

instructions. Extensive experiments demonstrate the effectiveness of GeoCAD in generation quality, validity and text-to-CAD consistency. Code will be available at `https://github.com/Zhanwei-Z/GeoCAD`.

# 1 Introduction

Computer-Aided Design (CAD) is pivotal in industrial design, driving innovation and efficiency across diverse domains such as mechanical manufacturing [8, 14, 17]. In CAD tools (such as SolidWorks and AutoCAD), the sketch-extrude-modeling (SEM) workflow [35, 45, 50, 48] is commonly employed, enabling users to control the parametric design process effectively. During this process, users sequentially extrude each 2D sketch into 3D shapes to construct complex solid CAD models, with each sketch comprising one or multiple local loops. Each local loop typically represents a pattern or geometric shape, serving as the fundamental closed-path element of a CAD model [50, 48].

In practice, any minor mistake in local parts (*i.e.*, local loops[3]) of a CAD model can potentially result in significant systemic errors. Thus, after drawing a draft CAD model, users generally need to modify its local parts to ensure that the final CAD product meets the expected functional or aesthetic requirements. Compared to manual modifications, if a deep-learning method can automatically adjust the shapes of local parts according to user-defined geometric instructions [4] (*e.g.*, an isosceles right triangle or a rectangular shape with one corner removed), it would significantly reduce labor costs in CAD product optimization. Moreover, such a method must retain the remaining CAD parts unchanged while ensuring that the newly generated local parts integrate with them without conflict. We refer to these capabilities as *local geometry-controllable CAD generation*.

Unfortunately, existing controllable CAD generation methods face challenges in achieving local geometry-control. Specifically, [50, 48, 46, 23, 57] typically take partial CAD parts or attributes (*e.g.*, incomplete sketches, topological or geometric parameters) as input and automatically generate new CAD models. Yet, they lack the ability to follow textual instructions, which hinders users from expressing their requirements intuitively and conveniently. To resolve this limitation, some text-to-CAD methods based on LLMs or transformers [41] have demonstrated meaningful progress [24, 18, 47, 55, 44, 43, 2, 61]. However, these methods are not applicable for local geometry-controllable generation. Specifically, [24, 18, 47, 44, 43, 2] typically generate a new CAD model from scratch based on textual instructions, making it difficult to fully focus on the required local parts. In addition, [18, 55, 43, 44] primarily collect textual descriptions of CAD models from global 3D views rather than local 2D views. These 3D views are generally oblique, which prevents capturing accurate geometric attributes (such as length and angle) of local parts for training. [61] can focus on local parts well but incorporates little geometric constraint, thereby struggling to follow geometric instructions.

In this paper, we propose GeoCAD, a user-friendly and local geometry-controllable CAD generation method. As shown in Fig. 1, GeoCAD takes the original CAD model, the local parts (highlighted in blue), and user-specific geometric instructions as inputs. The local parts are then generated by GeoCAD to align with the instructions, and are combined with the remaining parts to create new CAD models. To achieve this objective, the primary challenge is addressing the insufficiency of training data, specifically the geometric instructions for local parts. Given that manual captioning is prohibitively costly and labor-intensive, we introduce a complementary captioning strategy. Specifically, we categorize local parts into simple and complex groups based on their internal side types and numbers. Simple parts correspond to common geometric shapes (*e.g.*, triangles with three lines, quadrilaterals with four lines), while complex parts typically represent more intricate visual patterns. For complex parts, we render them as 2D images and then employ advanced vision large language models (VLLMs) [1, 4] to derive descriptive captions. However, for simple parts, VLLMs fail to achieve accurate fine-grained captioning. For example, VLLMs do not reliably distinguish whether a quadrilateral is a rhombus based solely on an image. To overcome this limitation, we introduce a vertex-based captioning method for simple parts. This involves extracting vertex coordinates from the original CAD model and then analyzing geometric attributes for accurate classification. For instance, if a quadrilateral has four lines of equal length, it is a rhombus; if it contains right angles, it is further categorized as a square. Utilizing the complementary strategy, we have successfully captioned approximately 221k different local parts, comprising 116k complex parts and 105k simple

---

[3]In the following, local parts refer to local loops, the finest-grained closed-path elements of a sketch.
[4]In this paper, geometric instructions denote textual captions of loop shapes.

parts. Inspired by the success of LLMs in text-to-CAD generation [47, 2, 55, 44, 43, 61], during training, given a CAD model, we randomly mask a local part and prompt LLMs to predict this part using the corresponding geometric instruction and the remaining visible parts as inputs. Once trained, in real-world applications, users can mask any local part for modification based on various geometric instructions. The new local parts generated by GeoCAD are then integrated with the remaining parts of the original CAD model to form new CAD models. Overall, our contributions are:

- We propose GeoCAD, a local geometry-controllable CAD generation method, enabling users to express design intent for specific parts through geometric instructions.
- To the best of our knowledge, GeoCAD is the first to achieve local geometry-control in the CAD generation field. To achieve this, we propose a complementary captioning pipeline to annotate ~221k distinct local parts for the following two-stage LLM fine-tuning.
- Extensive experiments demonstrate that GeoCAD significantly enhances generation quality, validity, and text-to-CAD consistency in local geometry-controllable CAD generation.

## 2 Related Work

**CAD Model Generation.** Existing CAD generation methods can be categorized into three types: constructive solid geometry (CSG), boundary representation (B-rep) and sketch-and-extrude modeling (SEM). CSG constructs CAD models by combining primitives (*e.g.*, cubes or spheres) into a tree [21, 6, 53, 34]. B-rep denotes CAD models as interconnected faces, edges, and vertices [3, 7, 49, 37]. Compared to CSG and B-rep, SEM-based methods [45, 50, 48, 17, 31, 55, 43, 44, 23, 61, 57, 5] are consistent with prevailing CAD tools, allowing users to sequentially extrude sketches into 3D shapes, with each sketch comprising one or multiple loops. Notably, within a sketch, any loop nested inside another loop serves as a hole. Recently, SEM-based controllable CAD generation has garnered a lot of attention due to its potential to revolutionize the design process [50, 48, 46, 23, 57]. Specifically, these methods allow for some level of control over the parts or attributes of the original CAD models. Among them, [50, 46, 23, 57] achieve sketch-level control, while [48] offers finer-grained control over local loops. Despite these capabilities, these methods struggle to follow textual instructions, limiting users from conveying their design intent in an intuitive and convenient manner.

On the other hand, current text-to-CAD methods that have demonstrated meaningful progress [24, 18, 47, 55, 44, 43, 2]. Notably, [24, 18, 47, 44, 43, 2] typically generate a new CAD model from the ground up based on textual instructions, which limits their ability to precisely target or refine specific local parts as per user specifications. Moreover, [18, 55, 43] primarily gather textual descriptions from global 3D perspectives rather than localized 2D views. These 3D perspectives are typically captured in oblique orientations, which limits their ability to precisely quantify critical geometric attributes (*e.g.*, length and angle) of local parts during the training process. [61] can effectively concentrate on the generation of local parts but fails to follow geometric instructions. To sum up, current studies lack the ability to achieve local geometry-controllable generation.

**Large Language Models (LLMs).** Compared to traditional deep-learning based models [62, 60, 9, 12, 36, 25], LLMs have recently demonstrated a remarkable ability to follow textual instructions [40, 1, 54, 51, 28, 10, 29]. Leveraging this capability, LLMs have shown notable versatility and efficacy across diverse applications [58, 22, 52, 13, 59, 38]. Users can employ various textual instructions to direct LLMs in accomplishing diverse tasks like code generation [15, 11] and question answering [39, 20]. As a branch, vision large language models (VLLMs) have also achieved significant success in vision domains [27, 56, 26]. More recently, both LLMs and VLLMs have shown promise in CAD generation [47, 2, 55, 44, 43, 61]. Specifically, [47, 55, 43, 44, 61] primarily rely on VLLMs for CAD caption synthesis or fine-tune LLMs or transformers [41] to generate CAD models from scratch. On the other hand, [2] employs a training-free manner to generate CAD codes via informative prompts. As mentioned above, these methods either cannot effectively focus on local generation or struggle to follow geometric instructions accurately. Distinguished from them, our GeoCAD excels in local part generation while precisely adhering to geometric instructions.

## 3 Methodology

In this section, we present GeoCAD, a user-friendly and local geometry-controllable CAD generation method. As shown in Fig. 1, GeoCAD incorporates three inputs: (1) an original CAD model,

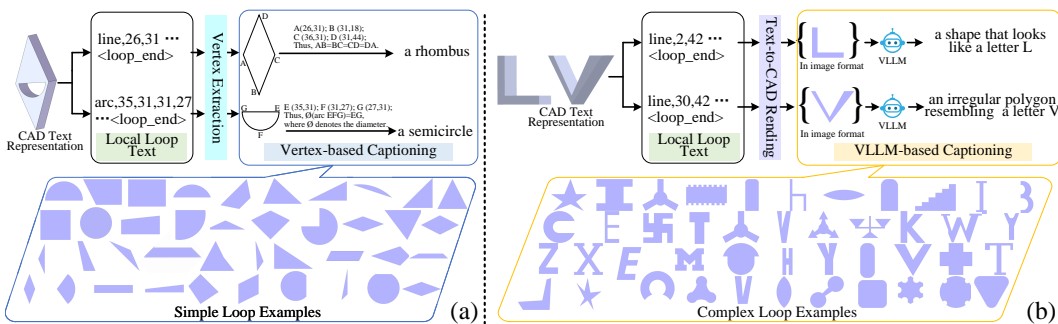

Figure 2: The complementary captioning strategy. (a) Vertex-based captioning for simple local parts. Vertex coordinates are initially extracted, followed by geometric analysis to enable precise captions. (b) VLLM-based captioning for complex local parts. We first convert complex parts into 2D images and subsequently employ powerful VLLMs to produce descriptive captions.

represented in a hierarchically textual format proposed by FlexCAD [61], (2) the local part designated for modification, and (3) geometric instructions specified by the user. GeoCAD then generates new CAD models, altering only the designated local part while closely adhering to the provided instructions. To achieve this, we first propose a complementary captioning strategy to generate ∼221k geometric instructions for local parts (Sec. 3.1). Building on these instructions, we then formulate a two-stage training pipeline to fine-tune LLMs for local CAD generation (Sec. 3.2).

## 3.1 Complementary Captioning for Local Parts

The main challenge in achieving local geometry-control is tackling the lack of training data, particularly concerning geometric instructions for local parts within 3D CAD models. Since manual captioning is excessively expensive and labor-intensive, we propose a complementary captioning strategy. In the beginning, we collect local parts (*i.e.*, local loops) from the CAD models within the DeepCAD dataset [45], filtering out duplicates and discarding invalid ones (*i.e.*, those that are not closed loops or involve intersecting line segments). Subsequently, we adopt the textual format introduced in FlexCAD [61] to represent CAD models and their local parts, where each local part is denoted as a contiguous string comprising the side type and vertex coordinates, as illustrated in Fig. 2. These local parts are then categorized into simple and complex groups based on their internal side numbers and types. Specifically, as shown in the lower part of Fig. 2(a), simple parts represent common geometric shapes (*e.g.*, triangles with three lines, quadrilaterals with four lines, sectors with two lines and an arc), making up roughly 50% of the entire set of local parts, while complex parts typically exhibit more intricate visual patterns as shown in the lower part of Fig. 2(b).

As shown in the upper part of Fig. 2(b), for complex parts, we transform them into 2D images and then leverage powerful VLLMs [1, 4] to obtain their geometric instructions (see the detailed prompts to guide VLLMs in the appendix). However, VLLMs exhibit limitations in fine-grained geometric descriptions for simple parts. For instance, they struggle to reliably discern whether a quadrilateral is a rhombus according to an image alone. To address this problem, we propose a vertex-based captioning method for simple parts. As shown in the upper part of Fig. 2(a), we first extract vertex coordinates from the original CAD text representation and then analyze geometric properties to precisely categorize these parts. For instance, given a quadrilateral, we can calculate its side lengths and inter-side angles based on its vertex coordinates. If it has four lines of equal length, it is a rhombus; if it includes right angles, it is further categorized as a square. Moreover, for partial simple parts, we also incorporate key dimensional parameters into the captions (such as the radius length of a circle and the side length of a square). In total, we annotate nearly 221k distinct local parts, consisting of 116k complex parts and 105k simple parts.

## 3.2 Fine-tuning LLMs with Geometric Instruction

With the geometric instructions derived in Sec. 3.1, we fine-tune LLMs to achieve local geometry-controllable CAD generation. The training procedure comprises two stages:

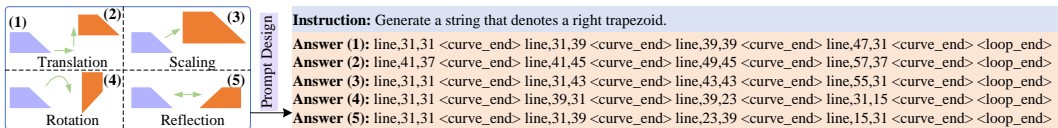

Figure 3: The prompt template used in stage 1. Local parts are first augmented through translation, scaling, rotation, and reflection. Subsequently, we construct the corresponding prompt that incorporates the geometric instruction, and ask LLMs to predict both the initial and augmented parts.

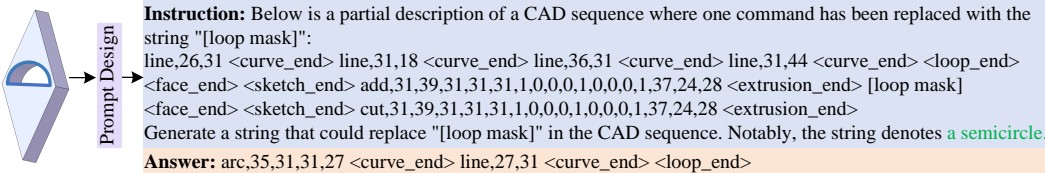

Figure 4: The prompt template used in stage 2. Given a local part (highlighted in blue) in a CAD model, we formulate the prompt that integrates the geometric instruction (highlighted in green) and the remaining parts of the CAD model, and require LLMs to predict this local part.

**Stage 1: Pre-training for CAD-text Alignment (Optional).** As mentioned in Sec. 3.1, we follow [61] to represent local parts using their internal side types and vertex coordinates. Since such CAD-specific geometric representation is typically absent from the pretraining corpus of LLMs, this stage focuses on aligning the representation of local parts with textual geometric instructions, thereby further enhancing the LLMs' understanding of the CAD-specific representation. Specifically, as illustrated in Fig. 3, for each local part, we apply random data augmentation via translation, scaling, rotation, and reflection. Notably, the geometric instructions of augmented samples remain unchanged due to the geometric consistency (*e.g.*, the geometric instructions of the augmented samples in Fig. 3 are all right trapezoids). Subsequently, for the initial and augmented samples, their corresponding instructions and answers are all employed to fine-tune LLMs.

**Stage 2: Instruction Fine-Tuning for Local Geometry-Control.** In practice, when modifying a specific part of a CAD model, it is crucial to retain the other parts of the CAD model unchanged. Additionally, the newly generated part should integrate with them without any conflicts. To this end, inspired by FlexCAD [61], at each epoch, for a given CAD model, we randomly mask a local part and design geometric instructions. These instructions are employed to prompt LLMs to predict this masked part autoregressively. However, FlexCAD's training process has one critical limitation: its prompts lack geometric constraints during training. Consequently, once trained, FlexCAD struggles to follow geometric instructions. In light of this, as shown in Fig. 4, our prompts incorporate the geometric instructions as constraints when fine-tuning LLMs to generate predictions. As shown in Fig. 5, during stages 1 and 2, the cross-entropy (CE) loss between the predicted tokens and the answer

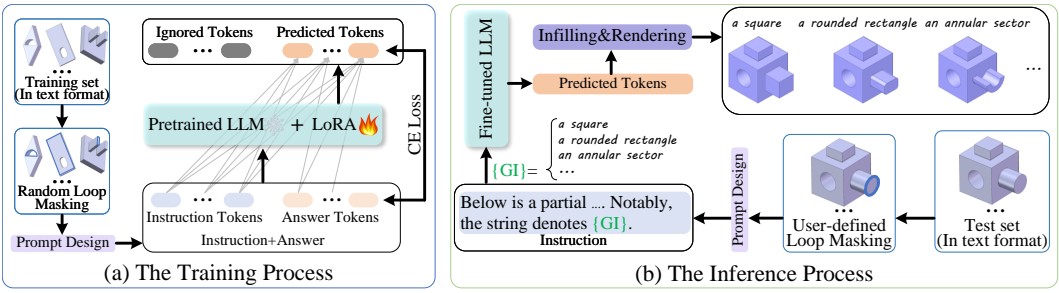

Figure 5: The overall framework of GeoCAD. (a) Training process. Given a CAD model, we randomly mask a local loop within it. During stages 1 and 2, we design the corresponding prompts (as introduced in Fig. 3 and Fig. 4), and fine-tune LLMs. (b) Inference process. Users can optionally mask any local part for modification, driven by various geometric instructions (GI). The mask part is then infilled with the predicted local parts to construct new CAD models.

tokens is back-propagated to update the trainable parameters of LLMs. Furthermore, we follow FelxCAD [61] to fine-tune the LLM using LoRA [16], which enables partial parameters training while freezing most parameter weights. This strategy allows us to retain the advantages of large-scale pre-trained models while accelerating convergence during optimization.

**Inference.** In practical applications, users can selectively mask any local part for modification, guided by various geometric instructions (*e.g.*, a square, a rounded rectangle, or an annular sector). The mask part is then replaced with the predicted local parts, which are seamlessly integrated with the remaining parts of the original CAD model to form new CAD models, as shown in Fig. 5(b).

## 4 Experiments

### 4.1 Experimental Setup

**Datasets.** To maintain consistency with prior research [61], we evaluate our GeoCAD on DeepCAD [45], a large-scale 3D sketch-extrude-modeling CAD dataset. This dataset contains 178,238 sketch-extrusion sequences, which are randomly partitioned into training, validation, and test subsets at a 90%-5%-5% ratio. Following established preprocessing protocols from SkexGen [50], we first eliminate duplicate and invalid sequences to ensure data quality. Subsequently, we follow FlexCAD [61] to convert the remaining CAD sequences into concise textual representations, which can be easily fed into LLMs. Within this dataset, we systematically collect and caption approximately 221k distinct local parts, including 116k complex parts and 105k simple parts.

**Implementation Details.** To ensure a fair comparison with FlexCAD [61], we adopt Llama-3-8B [32] as the base LLM, which achieves competitive performance among open-source LLMs. We use the same LoRA [16] setting as used in [61], with a rank of 8 and an alpha of 32. In stage 1, we implement translation, scaling, rotation, and reflection for simple parts, while applying only translation and scaling to complex parts to avoid semantic inconsistencies in captions. The model is trained on 8 A100 GPUs using AdamW [30], with a batch size of 32, a cosine annealing learning rate initialized at $5 \times 10^{-4}$, and trained for 10 and 30 epochs across stage 1 and stage 2. During inference, we set the temperature $\tau$ and Top-p at 0.9 and 0.9 to balance quality and validity in local generation.

**Metrics.** As this work pioneers local geometry-controllable CAD generation, we propose a comprehensive evaluation benchmark based on three key aspects:
1) Generation quality. We adopt metrics from prior work [50, 48, 61]. Specifically, *Coverage (COV)* measures the diversity of generated shapes and helps identify whether the model suffers from mode collapse. *Minimum Matching Distance (MMD)* reports the average minimum distance between real data and the generated set. *Jensen-Shannon Divergence (JSD)* quantifies the similarity between the distributions of real and generated samples. Together, these metrics measure generation quality on generated CAD models with respect to the test set.
2) Validity. Predicted local parts must form closed loops and must not contain intersecting line segments. In addition, these parts should seamlessly integrate with the existing parts to enable successful rendering into valid 3D shapes, rather than invalid or empty outputs. Following [61], we use *Prediction Validity (PV)* to quantify the overall validity rate of the generated predictions.
3) Text-to-CAD consistency. The generated 2D local parts should be consistent with user-defined geometric instructions. To measure this, we propose a *vertex-based score (Ver-score)* for assessing simple parts, and a *VLLM-based score (VLLM-score)* to evaluate complex parts. Finally, *Realism* denotes the human evaluation score, manually assessing whether the generated 3D CAD models fully satisfy user requirements for local geometry-control. See details of these metrics in the appendix.

### 4.2 Performance Comparision with Existing Methods

**Baselines.** As discussed above, most controllable CAD generation methods are not applicable to the local geometry-control task. Thus, we compare our GeoCAD with OpenAI-o3 [33], one of the most powerful closed-source LLMs, and FlexCAD [61], a state-of-the-art baseline for local CAD generation by fine-tuning LLMs. Without fine-tuning, the output format of the vanilla OpenAI-o3 model does not conform to the textual representation defined in [61], making it unable to directly generate local parts. To address this, we improve the performance of OpenAI-o3 with a few-shot learning strategy. Moreover, we manually enhance FlexCAD's performance when generating simple

Table 1: Performance comparison on the DeepCAD test set. Five-shot denotes that each prompt includes five exemplars selected from the training set that are either identical or semantically similar to the target instruction. Exemplars used in OpenAI-o3 consist of instructions and answers, following the format shown in Fig. 4. Best performances are in **bold**, and the second-bests are marked by *.

| Model | COV↑ | MMD↓ | JSD↓ | PV↑ | Ver-score↑ | VLLM-score↑ | Realism↑ |
|---|---|---|---|---|---|---|---|
| OpenAI-o3 (five-shot) | 53.6% | 1.64 | 1.49 | 65.7% | 33.6% | 22.1% | 18.7% |
| FlexCAD | 58.3% | 1.40 | 1.58 | 86.7% | 19.8% | 6.93% | 13.6% |
| FlexCAD (five-shot) | 59.4% | 1.37 | 1.34 | 88.1% | 43.5% | 26.8% | 20.2% |
| GeoCAD | 64.9%* | **1.13** | 0.98* | 90.5%* | 76.4%* | 65.7%* | 40.9%* |
| GeoCAD (five-shot) | **66.0%** | *1.16 | **0.80** | **92.3%** | **82.2%** | **68.2%** | **43.6%** |

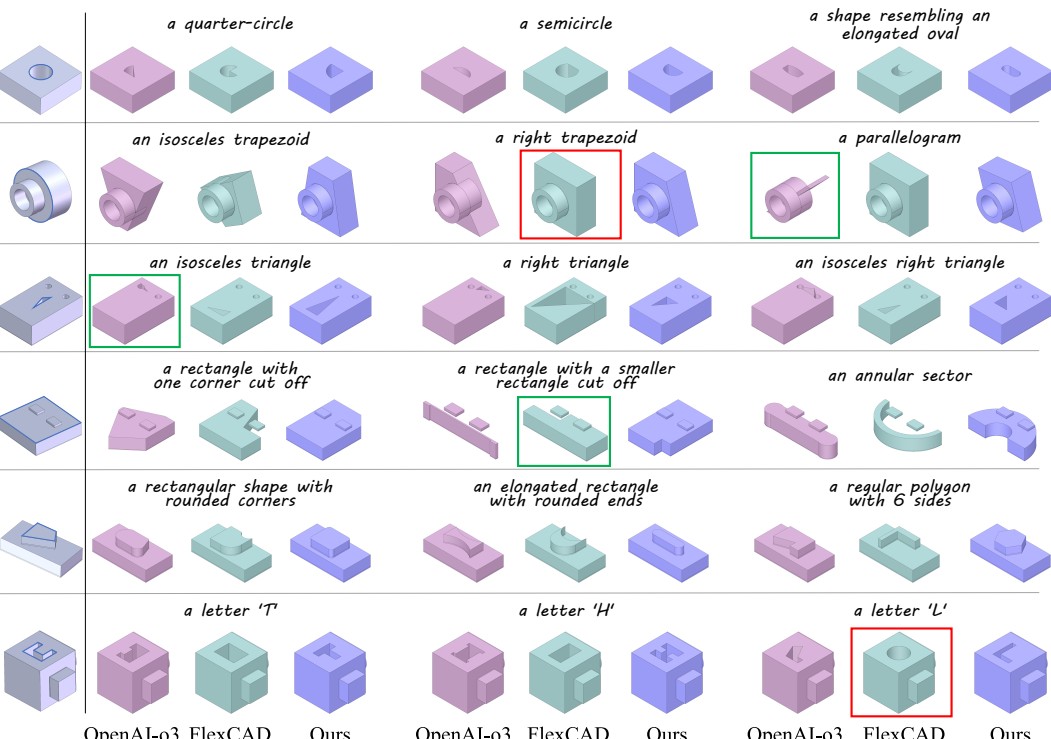

Figure 6: Qualitative comparison results for three methods. On the left, we show the original CAD models (in textual format), with the local parts to be modified (highlighted in blue, the same below). On the right, the upper section presents the user-defined geometric instructions, and the lower section displays the corresponding newly generated CAD models. Both FlexCAD and OpenAI-o3 are enhanced using five-shot learning. Red boxes indicate frequently occurring shapes in the training set (*e.g.*, circles or rectangles) that do not conform to the given geometric instructions. Green boxes highlight local parts that are poorly integrated with the remaining parts of the original CAD models.

parts by adjusting the internal curve types and numbers. For example, when aiming to generate an isosceles trapezoid, we try our best to guide FlexCAD to produce a loop composed of four lines.

**Quantitative Results.** We randomly sampled 1k CAD models from the test set. For each CAD model, a local part was randomly masked, and each method was prompted to generate 10 new parts using 5 simple and 5 complex geometric instructions. Here, simple and complex instructions correspond to the generation of simple and complex local parts, respectively. After infilling, this process yielded a total of 10k generated CAD models per method. To compute the COV, MMD, and JSD metrics, which rely on a subset of ground-truth samples, we randomly selected 3k CAD models from the test set and calculated the average results over three separate runs. As presented in Table 1, OpenAI-o3 delivers subpar performance without fine-tuning, even when supported by five-shot

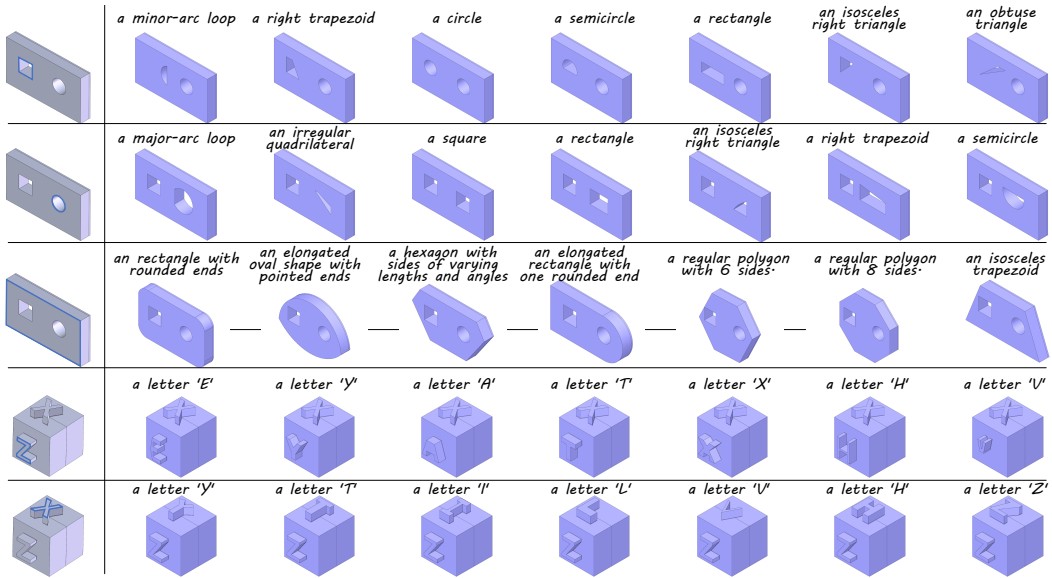

Figure 7: Additional qualitative results for GeoCAD. On the right, the upper section shows the user-defined instructions, while the lower section presents the newly generated CAD models.

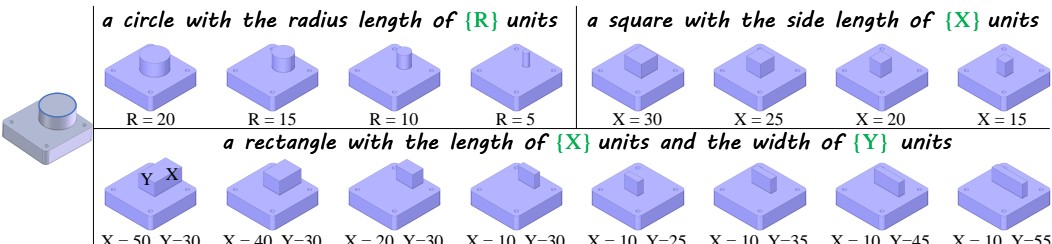

Figure 8: GeoCAD is capable of precisely controlling the key dimensional parameters. The right side displays the newly generated CAD models and the corresponding geometric instructions.

learning. In comparison, our proposed GeoCAD achieves superior results over the state-of-the-art baseline, FlexCAD, particularly in terms of Ver-score, VLLM-score, and Realism, with significant improvements of up to 38.7%, 41.4%, and 23.4%, respectively. This is mainly because FlexCAD lacks the ability to align with geometric instructions during the generation of local parts. On the other hand, the few-shot learning ability of LLMs leads to performance improvements for both FlexCAD and our GeoCAD. Overall, the results demonstrate the clear advantage of our GeoCAD in generation quality, validity, and text-to-CAD consistency.

**Qualitative Results.** To intuitively demonstrate performance, we randomly selected six CAD models from the test set. As shown in Fig. 6, the results clearly highlight that our GeoCAD significantly improves controllability and text-to-CAD consistency compared to existing baseline methods. In particular, GeoCAD is able to modify local parts in a way that closely adheres to user-defined geometric instructions. In contrast, FlexCAD struggles to comply with such instructions and frequently generates overly common shapes, such as circles or rectangles (see green boxes in Fig. 6). Moreover, as shown in the red boxes in Fig. 6, both OpenAI-o3 and FlexCAD often produce local parts that fail to align properly with the remaining parts of the original CAD models, resulting in outputs that are functionally or aesthetically implausible. These visualizations further validate the superior local controllability and effectiveness of our proposed GeoCAD.

Furthermore, we provide additional qualitative results generated by GeoCAD. As illustrated in Fig. 7, given a CAD model, GeoCAD is capable of effectively modifying any target loop within it to form simple or complex patterns. Moreover, for certain simple parts, we incorporate specific dimensional constraints into the instructions, such as the radius of a circle, the side length of a square, and the

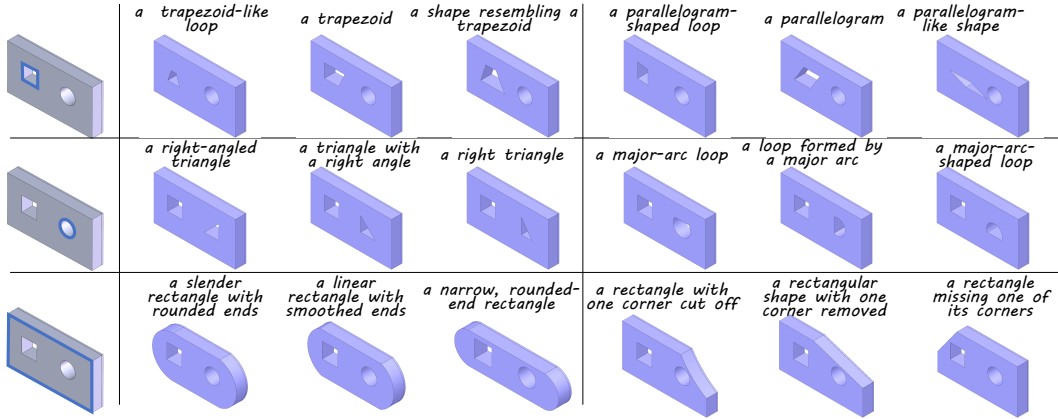

Figure 9: Generalization ability of GeoCAD. On the right, each row contains two groups, with each group comprising three examples generated based on semantically similar instructions.

Table 2: Effectiveness analysis of the complementary captioning strategy and pre-training. Only Vertex-based Captioning and Only VLLM-based Captioning indicate that local parts are described using only vertex-based or VLLM-based captioning, respectively. w/o stage 1 means that stage 1 is skipped, *i.e.*, no pre-training is conducted for aligning CAD data with textual descriptions. w/o data augmentation represents that only the original samples are used during pre-training, without any augmented data. Best performances are in **bold**.

| Model | COV↑ | MMD↓ | JSD↓ | PV↑ | Ver-score↑ | VLLM-score↑ |
|---|---|---|---|---|---|---|
| Only Vertex-based Captioning | 63.6% | 1.18 | 1.02 | 89.5% | **78.3%** | - |
| Only VLLM-based Captioning | 61.8% | 1.26 | 1.05 | 89.1% | - | 64.2% |
| w/o stage 1 | 61.3% | 1.21 | 1.16 | 89.6% | 71.5% | 60.4% |
| w/o data augmentation | 62.9% | 1.18 | 1.09 | 88.5% | 73.2% | 61.8% |
| Ours | **64.9%** | **1.13** | **0.98** | **90.5%** | 76.4% | **65.7%** |

length and width of a rectangle. As shown in Fig. 8, GeoCAD not only accurately generates the desired shapes but also adheres closely to the specified dimensional parameters. On the other hand, as shown in Fig. 9, GeoCAD demonstrates robust generalization capabilities in accurately understanding and executing semantically similar instructions, even when some of these instructions (*e.g.*, a narrow, rounded-end rectangle and a right-angled triangle) never appeared in the training data.

### 4.3 Ablation Studies

We conduct a series of ablation studies under the same experimental settings described in Table 2.

**Effectiveness of the complementary captioning strategy.** As shown in Table 2, using only vertex-based or VLLM-based captioning fails to generate complex parts (*e.g.*, a letter V) or simple parts (*e.g.*, a trapezoid), thereby failing to obtain the corresponding Ver-score and VLLM-score. In contrast, the complementary captioning integrating both of them leads to improved performance.

**Effectiveness of Pre-training.** As depicted in Table 2, omitting stage 1 results in the poorest performance, demonstrating that pre-training is essential for achieving preliminary text-CAD alignment. Additionally, excluding data augmentation during pre-training leads to a performance decline, indicating that diverse augmented samples enhance GeoCAD 's alignment capability. Together, these findings confirm the effectiveness of the pre-training process.

## 5 Conclusion

In this paper, we introduce a local geometry-controllable CAD generation method, GeoCAD, enabling users to specify design intent for specific parts through geometric instructions. To the best of our

knowledge, GeoCAD is the first to achieve local geometry-control in the CAD generation field. To accomplish this, GeoCAD introduces both vertex-based and VLLM-based captioning pipelines and employs a two-stage training strategy for LLM fine-tuning. Extensive qualitative and quantitative evaluations demonstrate that GeoCAD substantially improves generation quality, validity, and text-to-CAD consistency in local geometry-controllable CAD generation.

## Acknowledgement

This work was supported in part by the Key R&D Program of Zhejiang Province (2025C01212), in part by Yongjiang Talent Introduction Programme (2022A-240-G), in part by Ningbo Key R&D Program (2023Z229), in part by The National Nature Science Foundation of China (Grant NOs: 62273303, 62303406, 62273302, 62036009), in part by Ningbo Key R&D Program (NO.: 2025Z055).

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

# Appendix

Due to space limitations in the main paper, we provide additional results and discussions in this appendix, organized as follows:

- Sec. A: More Details about VLLM-based Captioning.
- Sec. B: Detailed Comparison with Existing Work.
- Sec. C: Detailed Categories of Simple Parts and Complex Parts.
- Sec. D: Details about Metrics for Evaluating Text-to-CAD consistency.
- Sec. E: LLMs of Different Scales.
- Sec. F: Sensitivity Analysis of Key Hyper-parameters in Sampling.
- Sec. G: Criteria for Dataset Selection.
- Sec. H: Sketch-level Editing.
- Sec. I: Failure Cases, Limitations and Future Work.
- Sec. J: Five-shot Prompt Example.
- Sec. K: Additional Qualitative Results.

## A    More Details about VLLM-based Captioning

The prompt used for VLLM-based captioning is as follows:

"Given a loop in a CAD sketch, provide a brief description of its geometric shape starting with 'a' or 'an' if identifiable; otherwise, state 'None'."

Using this prompt, we randomly caption 1k complex local parts with GPT-4o [1] and Qwen2.5-VL-72B-Instruct [4], respectively. Regardless of whether these models output a specific shape or 'None', we manually evaluate each result by judging its correctness as either "Yes" or "No". The overall captioning accuracy across these 1,000 parts is 91.3% for Qwen2.5-VL-72B-Instruct and 86.5% for GPT-4o. These results indicate that Qwen2.5-VL-72B-Instruct outperforms GPT-4o in this captioning task, which is consistent with with the latest multimodal model leaderboard rankings. Furthermore, given the lower cost of Qwen2.5-VL-72B-Instruct, we use it to caption the remaining complex parts.

## B    Detailed Comparison with Existing Work

As mentioned in lines 36-48 in our main paper, existing work struggles to achieve local geometry-controllable CAD generation. Here, we further highlight the differences between CAD-Editor [55], FlexCAD [61] and our GeoCAD. CAD-Editor has difficulty focusing on local generation for two main reasons: 1) It may unintentionally modify the remaining parts, resulting in outputs that do not align with user requirements (as illustrated in the last example of Fig. 1 in the original CAD-Editor paper). 2) CAD-Editor fails to accurately obtain angle and length information, making it incapable of generating even simple parts, such as a right triangle, let alone an isosceles right triangle, as mentioned in line 45 of our main paper. FlexCAD, on the other hand, can focus on local parts but incorporates minimal geometric constraints, thereby struggling to follow geometric instructions. In particular, FlexCAD is unable to understand, let alone follow, simple or complex geometric instructions. This limitation is clearly demonstrated in Fig. 1 of our main paper.

## C    Detailed Categories of Simple Parts and Complex Parts

The categories of simple parts include acute triangle, right triangle, obtuse triangle, isosceles triangle, isosceles right triangle (Notably, equilateral triangles do not occur in the DeepCAD [45] dataset), quadrilateral, trapezoid, isosceles trapezoid, kite (Two pairs of adjacent sides equal), parallelogram, rectangle, rhombus, square, circle, semicircle, quarter-circle, three-quarter circle, major-arc loop (defined as containing an arc longer than a semicircle), minor-arc loop (defined as containing an arc shorter than a semicircle), and so on. The remaining local parts are classified as complex, exhibiting more intricate and diverse visual patterns.

Table 3: Ablation studies on fine-tuning LLMs with different scales. Llama-3-8B is the model used in our main paper to enable a fair comparison with FlexCAD [61]. Transformer-4M is a small Transformer-based [42] language model, with a total number of trainable parameters comparable to that of our model in the main paper when using LoRA. Llama-3-8B-Full denotes full-parameter fine-tuning. Llama-3-8B, Qwen2.5-3B-Instruct, and Qwen2.5-7B-Instruct are all fine-tuned using LoRA. The best results are shown in **bold**, and the second-best results are marked with ∗.

| Model | COV↑ | MMD↓ | JSD↓ | PV↑ | Ver-score↑ | VLLM-score↑ |
|---|---|---|---|---|---|---|
| Transformer-4M | 59.1% | 1.32 | 1.26 | 85.5% | 69.3% | 51.2% |
| Llama-3-8B-Full | 67.5%* | 1.02* | 1.06 | 89.7% | 78.9%* | 64.2% |
| Llama-3-8B | 64.9% | 1.13 | 0.98* | **90.5%** | 76.4% | 65.7%* |
| Qwen2.5-3B-Instruct | 65.8% | **1.01** | 1.10 | 87.4% | 74.2% | 64.9% |
| Qwen2.5-7B-Instruct | **68.7%** | 1.05 | **0.86** | 90.1%* | **79.8%** | **70.2%** |

## D    Details about Metrics for Evaluating Text-to-CAD consistency

As mentioned in lines 213–217 of our main paper, we employ *Ver-score*, *VLLM-score*, and *Realism* to comprehensively evaluate model performance in terms of text-to-CAD consistency. Specifically, to compute *Ver-score*, we extract vertex coordinates from the generated local parts and analyze their geometric attributes to determine whether they align with the given geometric instructions. To obtain *VLLM-score*, we first render the local parts into images and then prompt two of the most powerful VLLMs, GPT-4o [1] and Qwen2.5-VL-72B-Instruct [4], to judge whether the rendered images match the corresponding instructions, assigning a binary label: "Yes" or "No." We report the average of their scores in Table 1 of our main paper, where both models significantly outperform the baselines. To evaluate *Realism*, we randomly render 500 newly generated CAD models into images, with the modified local parts clearly marked. Five crowd workers are then asked to assess whether the generated local parts align with the geometric instructions and do not conflict with the remaining parts. If both criteria are satisfied, they assign a binary label: "Yes"; otherwise, "No." The average score from these workers is reported in Table 1 of our main paper.

## E    LLMs of Different Scales

As shown in Table 3, Transformer-4M achieves the lowest performance, confirming that LLMs play a key role in enhancing local CAD generation. Llama-3-8B-Full performs comparably to Llama-3-8B, demonstrating the effectiveness of the LoRA strategy [16]. As two of the most popular open-source LLMs, Qwen2.5-7B-Instruct slightly outperforms Llama-3-8B.

## F    Sensitivity Analysis of Key Hyper-parameters in Sampling

Table 4: Effectiveness analysis of key hyper-parameters, including the sampling temperature $\tau$ and Top-p. Best performances are in **bold** and the second-bests are marked by ∗.

| Model | COV↑ | MMD↓ | JSD↓ | PV↑ | Ver-score↑ | VLLM-score↑ |
|---|---|---|---|---|---|---|
| $\tau = 0.7$ | 63.4% | 1.18 | 1.03 | **91.2%** | 75.9% | 63.2% |
| $\tau = 0.9$ | 64.9%* | **1.13** | 0.98* | 90.5%* | 76.4%* | **65.7%** |
| $\tau = 1.1$ | **65.6%** | 1.16* | **0.95** | 89.1% | **77.5%** | 65.1%* |
| Top-p = 0.8 | 64.1% | 1.21 | 1.09 | **91.0%** | 75.3% | 64.4% |
| Top-p = 0.9 | 64.9%* | **1.13** | 0.98* | 90.5%* | 76.4%* | 65.7%* |
| Top-p = 1.0 | **65.2%** | 1.18* | **0.92** | 88.3% | **76.9%** | **66.8%** |

As shown in Table 4, we conduct a sensitivity analysis on key hyperparameters, including the sampling temperature $\tau$ and Top-p. All other experimental settings follow those described in Section 4.2 of our main paper. In general, increasing $\tau$ or Top-p results in more diverse and stochastic predictions. However, this comes at the cost of reduced PV, while other metrics tend to improve, consistent with

findings in [61]. In our experiments, we balance this trade-off by selecting $\tau$ and Top-p values that ensure the PV remains above 90%.

## G  Criteria for Dataset Selection

DeepCAD [45] is a suitable dataset for evaluation, and the reasons are detailed below: 1) Scale: Deep-CAD is a large-scale 3D CAD dataset, comprising over 178k samples. 2) Relevance to Controllability: Compared to 2D sketch datasets, DeepCAD better reflects the requirements of controllable generation, as aligning local parts within 3D CAD models is both more challenging and more practical. 3) Design Process Alignment: In contrast to other 3D CAD datasets, such as the ABC dataset [19], DeepCAD includes sketch-and-extrusion sequences that closely mirror the design workflows of commercial CAD tools like SolidWorks and AutoCAD. 4) Community Adoption: Due to its characteristics, DeepCAD is also the only choice for prior studies, including SkexGen [50], HNC-CAD [48], CAD-Editor [55], CADFusion [44], Text2CAD [18], and FlexCAD [61].

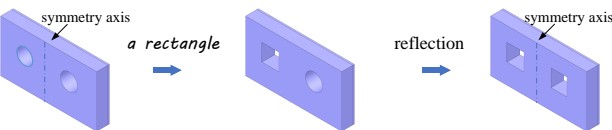

Figure A1: An example of sketch-level editing.

## H  Sketch-level Editing

For sketch-level editing, if a sketch contains multiple loops, ideally, we would like to learn the inter-loop constraints (*e.g.*, symmetry, patterns, etc.) that define the overall structure. However, as mentioned above, DeepCAD is currently the only dataset suitable for controllable 3D CAD generation, and unfortunately, such inter-loop constraint annotations are not provided in the dataset. Fortunately, even without supervision from these constraints, sketch-level editing is still achievable based on our loop-level editing capability. This is because the loop serves as the fundamental element of a sketch. For example, as shown in Fig. A1, if a user selects a sketch consisting of two symmetric loops and wishes to replace them with another pair of symmetric loops, the following automatic process can be performed: 1) Estimate the center point of each original loop by averaging its coordinate points, which are extracted using string matching. 2) Determine the symmetry axis based on the two center points. 3) Generate a new local loop through GeoCAD replacing one of the orignal loops. 4) Reflect the newly generated loop across the symmetry axis to produce the second symmetric loop, thereby replacing both original loops.

## I  Failure Cases, Limitations and Future Work

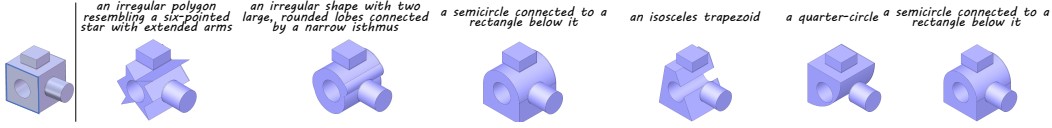

Figure A2: Failure cases. The generated local parts align well with the user's geometric instructions but do not integrate smoothly with the remaining parts of the original CAD model.

**Failure cases.** Despite notable advancements, our GeoCAD sometimes results in failure cases. As shown in Fig. A2, given a CAD model, when only the special part is modified (*i.e.*, the part upon which the remaining parts are constructed and strictly aligned in size), the unchanged remaining parts may lead to structural conflicts with it. To mitigate this issue, when modifying the special parts, users should provide geometric instructions that account for the constraints imposed by the remaining parts, since the DeepCAD dataset does not annotate the relationships between different parts.

**Limitations and future work.** In this paper, we fine-tune LLMs to enable local geometry-controllable CAD generation, primarily guided by textual instructions. However, in practice, certain complex local

parts may be difficult or even impossible to describe using text alone. Thus, in the future, if users can complement textual inputs with hand-drawn images for local geometry-controllable CAD generation, they may be able to convey their design intent more effectively. Given the strong capabilities of VLLMs in both CAD generation and text understanding, our future work aims to develop a more advanced multimodal LLM tailored for controllable CAD generation from both text and image inputs.

## J   Five-shot Prompt Example

To better illustrate the implementation details of the baselines and our GeoCAD in Table 1 of our main paper, we present a five-shot prompt example, as shown in Fig. A3.

## K   Additional Qualitative Results

We provide additional qualitative results in Fig. A4.

You answer questions about controllable CAD generation. When answering user questions, please follow these examples:

Example 1
Instruction:
Below is a partial description of a CAD sequence where one command has been replaced with the string "[loop mask]":
line,0,26 <curve_end> line,1,26 <curve_end> line,1,28 <curve_end> line,0,28 <curve_end> <loop_end> <face_end> [loop mask] <face_end>
<sketch_end> add,31,45,31,31,31,1,0,0,0,1,0,0,0,1,44,31,36 <extrusion_end>
Generate an string that could replace "[loop mask]" in the CAD sequence. Notably, the string denotes an isosceles right triangle.
Answer:
line,0,28 <curve_end> line,1,29 <curve_end> line,1,28 <curve_end> <loop_end>

Example 2
Instruction:
Below is a partial description of a CAD sequence where one command has been replaced with the string "[loop mask]":
line,42,47 <curve_end> line,43,46 <curve_end> line,43,47 <curve_end> <loop_end> <face_end> line,43,15 <curve_end> line,44,15
<curve_end> line,44,16 <curve_end> <loop_end> <face_end> [loop mask] <face_end> <sketch_end> add,28,34,31,31,31,1,0,0,0,0,1,0,-
1,0,38,46,31 <extrusion_end>
Generate an string that could replace "[loop mask]" in the CAD sequence. Notably, the string denotes an isosceles right triangle.
Answer:
line,43,47 <curve_end> line,44,46 <curve_end> line,44,47 <curve_end> <loop_end>

Example 3
Instruction:
Below is a partial description of a CAD sequence where one command has been replaced with the string "[loop mask]":
line,14,57 <curve_end> line,16,55 <curve_end> line,16,10 <curve_end> line,21,4 <curve_end> line,47,4 <curve_end> line,48,8 <curve_end>
line,48,8 <curve_end> line,47,5 <curve_end> line,21,5 <curve_end> line,16,10 <curve_end> line,16,55 <curve_end> line,14,58 <curve_end>
<loop_end> <face_end> <sketch_end> add,31,63,31,31,31,1,0,0,0,0,1,0,-1,0,21,39,45 <extrusion_end> line,26,0 <curve_end> line,29,0
<curve_end> line,36,7 <curve_end> line,36,62 <curve_end> line,36,62 <curve_end> line,36,7 <curve_end> line,29,0 <curve_end> line,26,0
<curve_end> <loop_end> <face_end> line,26,0 <curve_end> line,29,0 <curve_end> line,36,7 <curve_end> line,36,62 <curve_end> line,26,62
<curve_end> <loop_end> <face_end> [loop mask] <face_end> <sketch_end> cut,27,31,31,31,55,-1,0,0,0,0,1,0,1,0,16,29,17 <extrusion_end>
Generate an string that could replace "[loop mask]" in the CAD sequence. Notably, the string denotes an isosceles right triangle.
Answer:
line,29,0 <curve_end> line,36,0 <curve_end> line,36,7 <curve_end> <loop_end>

Example 4
Instruction:
Below is a partial description of a CAD sequence where one command has been replaced with the string "[loop mask]":
[loop mask] <face_end> <sketch_end> add,31,62,31,31,31,1,0,0,0,0,1,0,-1,0,36,31,44 <extrusion_end> line,5,14 <curve_end> line,5,31
<curve_end> line,22,48 <curve_end> line,40,48 <curve_end> line,57,31 <curve_end> line,57,14 <curve_end> line,31,40 <curve_end>
<loop_end> <face_end> <sketch_end> add,31,39,31,31,31,1,0,0,0,0,1,0,-1,0,39,31,48 <extrusion_end>
Generate an string that could replace "[loop mask]" in the CAD sequence. Notably, the string denotes an isosceles right triangle.
Answer:
line,31,17 <curve_end> line,59,17 <curve_end> line,31,45 <curve_end> <loop_end>

Example 5
Instruction:
Below is a partial description of a CAD sequence where one command has been replaced with the string "[loop mask]":
line,6,12 <curve_end> line,6,50 <curve_end> line,43,50 <curve_end> line,56,36 <curve_end> line,56,33 <curve_end> line,32,33 <curve_end>
line,32,29 <curve_end> line,56,29 <curve_end> line,56,26 <curve_end> line,43,12 <curve_end> <loop_end> <face_end> line,43,12
<curve_end> line,56,12 <curve_end> line,56,26 <curve_end> <loop_end> <face_end> line,43,50 <curve_end> line,56,36 <curve_end>
line,56,50 <curve_end> <loop_end> <face_end> <sketch_end> add,31,39,31,31,31,1,0,0,0,1,0,0,0,1,44,24,34 <extrusion_end> [loop mask]
<face_end> line,20,61 <curve_end> line,42,39 <curve_end> line,42,61 <curve_end> <loop_end> <face_end> <sketch_end>
cut,31,57,31,31,31,1,0,0,0,1,0,0,0,1,27,45,34 <extrusion_end> line,22,1 <curve_end> line,40,1 <curve_end> line,40,61 <curve_end> line,22,61
<curve_end> <loop_end> <face_end> <sketch_end> add,31,41,30,33,39,1,0,0,0,1,0,0,0,1,27,3,31 <extrusion_end> line,0,25 <curve_end>
line,6,37 <curve_end> line,56,37 <curve_end> line,62,25 <curve_end> <loop_end> <face_end> <sketch_end>
cut,5,31,11,33,44,0,1,0,0,0,1,1,0,0,19,31,34 <extrusion_end>
Generate an string that could replace "[loop mask]" in the CAD sequence. Notably, the string denotes an isosceles right triangle.
Answer:
line,20,1 <curve_end> line,42,1 <curve_end> line,42,23 <curve_end> <loop_end>

Instruction:
Below is a partial description of a CAD sequence where one command has been replaced with the string "[loop mask]":
line,4,20 <curve_end> line,22,14 <curve_end> line,47,14 <curve_end> line,58,25 <curve_end> line,47,25 <curve_end> line,22,25 <curve_end>
line,4,25 <curve_end> <loop_end> <face_end> line,4,37 <curve_end> line,22,37 <curve_end> line,22,48 <curve_end> line,4,42 <curve_end>
<loop_end> <face_end> line,22,37 <curve_end> line,47,37 <curve_end> line,47,48 <curve_end> line,22,48 <curve_end> <loop_end>
<face_end> arc,47,25,51,27 <curve_end> line,53,31 <curve_end> line,58,31 <curve_end> line,58,25 <curve_end> <loop_end> <face_end>
arc,47,37,51,35 <curve_end> line,53,31 <curve_end> line,58,31 <curve_end> line,58,37 <curve_end> <loop_end> <face_end> [loop mask]
<face_end> <sketch_end> add,29,33,31,31,31,1,0,0,0,1,0,0,0,1,32,18,31 <extrusion_end>
Generate an string that could replace "[loop mask]" in the CAD sequence . Notably, the string denotes an isosceles right triangle.
Answer:

Figure A3: A five-shot prompt example used in Table 1 of our main paper.

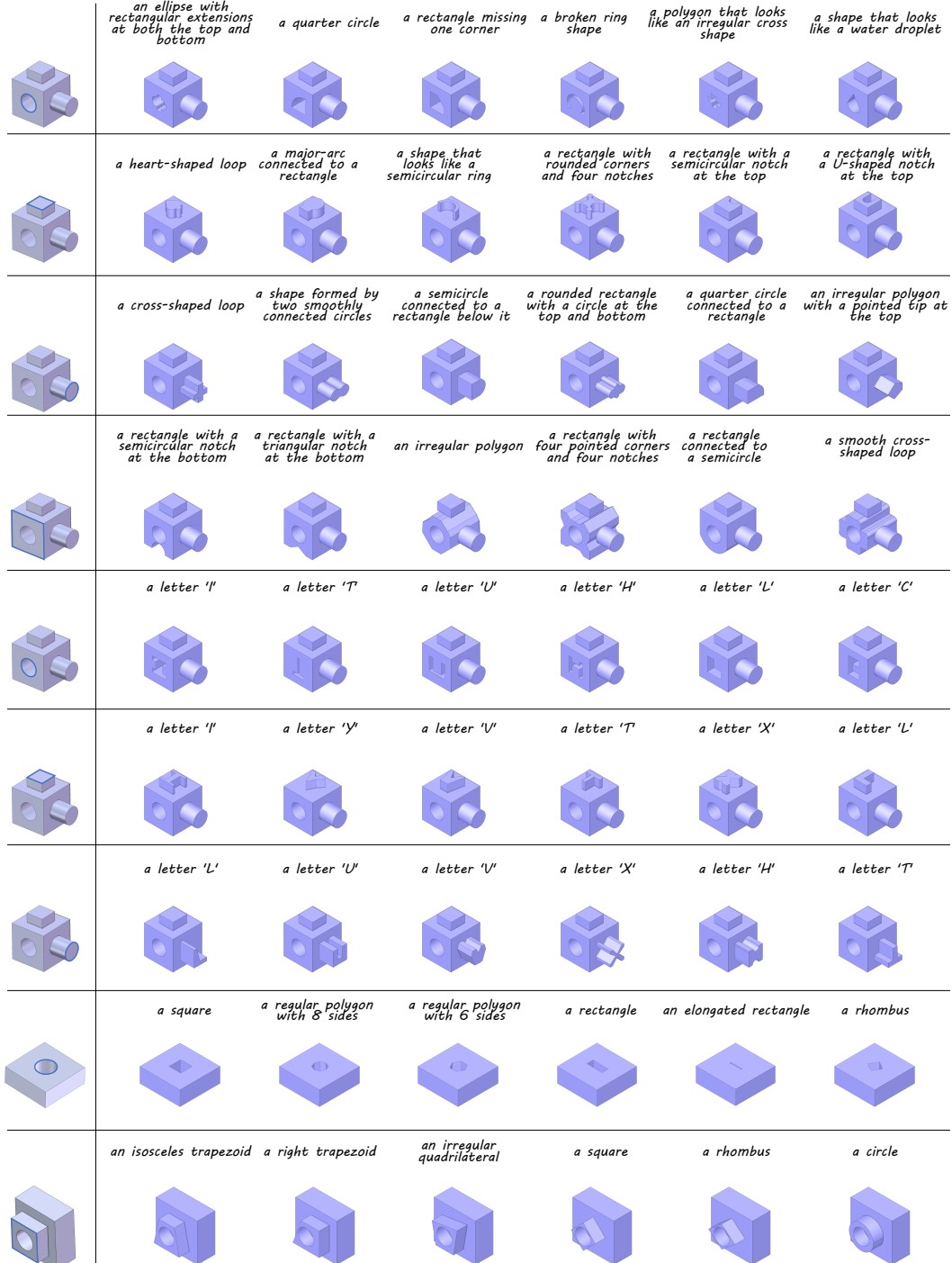

Figure A4: Additional Qualitative Results.

