# OpenReview forum: "GeoCAD: Local Geometry-Controllable CAD Generation with Large Language Models"
_NeurIPS.cc/2025/Conference — NeurIPS 2025 poster_

### Official Review · Reviewer_HFw7 · 2025-06-23

**Clarity:** 3
**Significance:** 3
**Originality:** 3
**Rating:** 5
**Confidence:** 1

**Summary:**

This paper proposes GeoCAD, a local geometry controllable generation method. It can take the original CAD model and user-specific geometric instructions  as inputs. It is one of the first model to achieve geometry control in the generation of CAD.  Extensive results shows that the proposed GeoCAD approach significantly enhances generation quality, validity, and text-to-CAD consistency in local geometry-controllable CAD generation.

**Questions:**

I wonder if the scaling law also happens here when evaluating the LLMs’ ability in predicting the masked local part. Will larger LLMs perform better on this CAD task after the proposed finetuning strategy?

**Ethical Concerns:**

["NO or VERY MINOR ethics concerns only"]

**Final Justification:**

The rebuttal addressed my concerns. It's good to see the human evaluation results indicating the quality of the local parts and large model also works. I will raise my score.

**Limitations:**

Yes.

**Paper Formatting Concerns:**

Figure 5 can be put at the top rather than at the bottom.

I would suggest the authors put more introductions about the evaluation metrics used. For example, currently it is unclear how Realism (human evaluation score) works from the main paper.

**Quality:**

3

**Strengths And Weaknesses:**

Strengths:

- The motivation of this paper sounds reasonable. GeoCAD can achieve local geometry-control in the CAD generation field.
- The two stage training pipeline is novel and effective. Different data augmentation strategies are used in stage 1.
- The paper is well written and easy to follow. Main figures in this paper are well drawn and make the method clear to understand.
- The experiments are done comprehensively and there are a lot of qualitative results given, demonstrating the superiority of the proposed GeoCAD method compared with other baselines such as OpenAI-o3 and FlexCAD.

Weaknesses:
- The authors claim that VLLMs exhibit limitations in fine-grained geometric descriptions for simple parts. For instance, they struggle to reliably discern whether a quadrilateral is a rhombus according to an image alone. But this is well supported. Actually state-of-the-art VLLMs can already perform well on existing benchmarks asking for geometric understanding.

- The quality of the annotated 221k local parts are not clear. Human evaluation for at least part of them is needed.

- The LLM is fine-tuned using LoRA, while it lacks comparisons on using other parameter efficient finetuning strategies as well as full-model finetuning. It is also not sure if the rank of 8 and an alpha of 3 is the optimum setting for the finetuning.

---

> ### Author Rebuttal · Authors · 2025-07-31
>
> **Q1: State-of-the-art VLLMs can already perform well on existing benchmarks asking for geometric understanding.**
> A1: Thanks for your valuable feedback!
> We agree that SOTA VLLMs (such as OpenAI-o3) can already perform well on geometric understanding.
> However, when annotating simple local parts, Vertex-based captioning offers two key advantages over VLLM-based captioning:
> 1) Cost-free: It does not require deploying any local models or invoking paid API services.
> 2) Fully accurate: Compared to VLLM-based captioning, rule-based captioning ensures 100% correctness in annotation by explicitly extracting angle and length information of loops.
>
> **Q2: The quality of the annotated 221k local parts are not clear. Human evaluation for at least part of them is needed.**
> A2: We are pleased to see that the reviewer also recognized this point.
> In fact, we have conducted a human evaluation of the annotated local parts in the supplementary material.
> Specifically, as described in lines 20–22 of the supplementary material, we manually assessed the correctness of each caption by labeling it as either "Yes" (correct) or "No" (incorrect).
> Based on this evaluation of 1,000 randomly sampled parts, the overall captioning accuracy is 91.3% for Qwen2.5-VL-72B-Instruct and 86.5% for GPT-4o.
>
> **Q3: Use other parameter efficient finetuning strategies as well as full-model finetuning. It is also not sure if the rank of 8 and an alpha of 3 is the optimum setting for the finetuning.**
> A3: In fact, we have compared LoRA with full-model finetuning in the Table 1 in the supplementary material.
> The results show that full-model finetuning performs comparably to LoRA-based tuning, demonstrating the effectiveness of the LoRA strategy.
> Here, we compare another parameter efficient finetuning strategy (i.e., QLoRA [1]) with LoRA.
> The results are as follows:
> |Method|COV$\uparrow$|MMD$\downarrow$|JSD$\downarrow$|PV$\uparrow$|Ver-score$\uparrow$|VLLM-score$\uparrow$|
> |-|-|-|-|-|-|-|
> |QLoRA|62.2%|1.17|1.01|85.5%|74.5%|63.8%|
> |LoRA (rank=8, alpha=64)|__65.7%__|1.10|0.92|89.7%|__77.8%__|66.0%|
> |LoRA (rank=16, alpha=32)|65.3%|__1.08__|__0.86__|__90.9%__|77.2%|__66.3%__|
> |LoRA (rank=8, alpha=32,ours)|64.9%|1.13|0.98|90.5%|76.4%|65.7%|
>
> The results show that QLoRA performs the worst among the evaluated methods.
> On the other hand, increasing the LoRA rank and alpha leads to slight performance improvements.
> However, higher rank and alpha values result in slower convergence.
> To balance performance and training efficiency and ensure a fair comparison with FlexCAD, we choose rank = 8 and alpha = 32 in our main experiments. Thanks!
>
> [1] Dettmers T, Pagnoni A, Holtzman A, et al. Qlora: Efficient finetuning of quantized llms. NeurIPS 2023.
>
> **Q4: Will larger LLMs perform better on this CAD task after the proposed finetuning strategy?**
> A4: Here, we compare the performance of Llama-3-70B equipped with LoRA.
> The results are as follows:
> |Method|COV$\uparrow$|MMD$\downarrow$|JSD$\downarrow$|PV$\uparrow$|Ver-score$\uparrow$|VLLM-score$\uparrow$|
> |-|-|-|-|-|-|-|
> |Llama-3-70B|__69.8%__|__0.95__|__0.83__|__92.2%__|__81.0%__|__71.6%__|
> |Llama-3-8B (ours)|64.9%|1.13|0.98|90.5%|76.4%|65.7%|
>
> The results show that Llama-3-70B achieves superior performance, but it is more time-consuming and costly, as also explored in FlexCAD. In other words, as the LLM scale increases, both generation quality and instruction-following ability improve accordingly. Thanks!

---

### Official Review · Reviewer_ZjoJ · 2025-07-02

**Clarity:** 3
**Significance:** 2
**Originality:** 1
**Rating:** 4
**Confidence:** 4

**Summary:**

This paper proposes a method to support user-friendly CAD model editing through prompts. Using prompts to edit CAD models is an interesting problem; this method additionally requires the user to select a local profile to be edited. The technical contributions focus on: 1. The captioning strategy - it creates a prompt-to-local-part dataset suitable for both simple and complex parts generation. 2. The pre-training strategy on the prompt-to-local-part dataset, which enables the VLLM to learn local parts generation better. The experiments show that the method can produce better results than several baselines and can follow prompts to generate desired local profiles.

**Questions:**

Is it possible to also use prompt language to change the extrusion length and the extrusion combination operations (union and differ)? It would be better to discuss this point.

**Ethical Concerns:**

["NO or VERY MINOR ethics concerns only"]

**Final Justification:**

The rebuttal addressed some of my concerns.

I still think the authors should add more examples to demonstrate the method can hanlde more complicated cases.
For example only editing part of the sketch and editing the symmetry sketches.

Overall, I remain borderline acceptance.

**Limitations:**

yes

**Quality:**

3

**Strengths And Weaknesses:**

Strengths:
1.The proposed problem is novel; using natural language to efficiently edit CAD models is an interesting problem.

2.The performance is impressive compared to several baselines.

3.The paper is well-structured and easy to understand. The experimental results are also promising.

Weaknesses:

1.The input requires a clean CAD shape program, which limits the application scope. Many CAD shapes in the ABC dataset cannot be edited by this method.

2.The editing operations have many limitations. Many local patterns are difficult to describe through text. Additionally, users can only edit the 2D sketch, not the extrusion length. The text prompt will change the entire 2D sketch selected by the user, not just part of the sketch. I don't see more efficiency or flexibility compared to manual editing of 2D sketches.

3.The authors proposed two main technical contributions - the separated captioning strategy and the pre-training - which appear very intuitive and specifically designed only for this task.

Overall, I feel the technical contributions are not significant, and the use case is not quite general. There are many limitations that need to be addressed for more practical use cases.

---

> ### Author Rebuttal · Authors · 2025-07-31
>
> **Q1:The input requires a clean CAD shape program, which limits the application scope. Many CAD shapes in the ABC dataset cannot be edited by this method.**
> A1: We appreciate the reviewer’s concern regarding the current scope of our method. We would like to clarify the following point:
> 1) Our work focuses on common and practical CAD primitives, specifically sketches and extrusions.
> They are sufficiently expressive to cover a wide variety of shapes, and form the basis of many real-world CAD models [1].
> 2) As mentioned in [1] and [2], the ABC dataset is provided in the B-rep format, with no sufficient information to recover the user's design process [1], making it unsuitable for controllable CAD generation [2].
> In contrast, we focus on the sketch-and-extrusion format, which align with the CAD design process in commercial CAD tools like SolidWorks and AutoCAD.
> 3) GeoCAD is designed to be extensible. Our GeoCAD can be easily adapted to support more complex operations such as chamfer and revolve once relevant data becomes available for training.
>
> [1] R. Wu, C. Xiao, and C. Zheng. Deepcad: A deep generative network for computer-aided design models. ICCV 2021.
> [2] Z. Zhang, S. Sun, W. Wang, et al. Flexcad: Unified and versatile controllable cad generation with fine-tuned large language models. ICLR, 2025.
>
> **Q2: Many local patterns are difficult to describe through text.**
> A2: We appreciate that the reviewer also recognized this point.
> In fact, we have discussed this in the limitation section.
> As shown in lines 88-92 in the supplementary material,
> in practice, certain complex local parts may be difficult or even impossible to describe precisely using text alone.
> As a potential direction for future work, we believe that combining textual input with hand-drawn images for
> local geometry could enable users to convey their design intent more effectively and accurately.
>
> **Q3: the entire sketch, the extrusion length and the extrusion combination operations (union and differ)**
> A3: This a good question. Actually, sketch-level editing is achievable based on our loop-level editing capability.
> This is because the loop serves as the fundamental element of a sketch. For example, if a user selects a sketch consisting of two symmetric trapezoids and
> wishes to replace them with two symmetric letter "E" shapes,
> the following automatic process can be performed:
> 1) Estimate the center point of each trapezoid by averaging its coordinate points, which are extracted using string matching.
> 2) Determine the symmetry axis based on the two center points.
> 3) Generate a local letter "E" through GeoCAD replacing one of two trapezoids.
> 4) Reflect the generated "E" across the symmetry axis to produce the second symmetric letter "E", thereby replacing both original trapezoids.
>
> Thus, for sketch-level editing, it is unnecessary for the model to generate the entire sketch; generating only the fundamental loop using GeoCAD is sufficient.
> Subsequently, the remaining loops within the same sketch can be derived from the fundamental loop through scaling, rotation, translation, reflection or pattern operations.
>
> On the other hand, there is no need to control parameters such as extrusion length based on designed prompts. Specifically:
> 1) Modifying the extrusion length and performing extrusion combination operations both require only a single parameter change in the CAD text. Instead of designing detailed prompts, directly modifying the parameters in the original CAD text is more efficient.
> 2) For more complex extrusion modifications, users must specify detailed parameters such as 3D translations and rotations.
> This process can be quite unintuitive for non-experts and unnecessarily tedious for experts.
> In contrast, GeoCAD is designed to be user-friendly for both non-experts and experts, where users only need to select the part they wish to modify and provide geometric instructions for local parts.
>
> **Q4:Two main technical contributions are very intuitive and specifically designed only for this task.**
> A4:
> Actually, the two-stage training process are common in LLM training. We are the first to propose a two-stage LLM training pipeline in controllable CAD generation.
> Specifically, in Stage 1, we introduce a data augmentation strategy tailored for local CAD generation, which was recognized by Reviewer HFw7 as both novel and effective.
> In Stage 2, we intentionally retain the standard training strategy with minimal modifications from FlexCAD.
> The reasons are as follows:
> i) To demonstrate that our method is simple yet effective. Without relying on elaborate training strategies, GeoCAD significantly outperforms FlexCAD and substantially enhances geometric instruction-following capabilities.
> ii) To facilitate the development of an omni CAD LLM. A standard and unified training strategy allows for better integration between different models, such as FlexCAD and our GeoCAD, bringing us closer to a general-purpose, omni-capable CAD foundation model.
>
> On the other hand, our paper presents novelty not only in the technical aspects but also in the task formulation.
> Specifically, we propose and solve a new task (i.e., local geometry-controllable CAD generation, see lines 27–35 in our main paper).
> **All reviewers recognized the value of our proposed task (Reviewer td3y: very sensible, Reviewer 6Hfr: more impactful than generating from scratch, Reviewer ZjoJ: novel and interesting, Reviewer HFw7: reasonable).**
> To support this task, we also collected a dedicated dataset of local parts and their corresponding captions, which Reviewer 6Hfr described as a sizeable and solid contribution.

---

> > ### Comment · Reviewer_ZjoJ · 2025-08-06
> >
> > Thanks for the authors' response.
> >
> > I agree that the extrusion length doesn't need to be changed by prompt.
> >
> > I still think the authors should add more examples to demonstrate the method can handle more complicated cases. For example only editing part of the sketch and editing only part of the symmetry sketches.
> >
> > Also strengthen the limitation discussion section for local patterns, chamfer and revolve operations, and generalization for other representation, not only limited to the clean program.
> >
> > Overall, I remain positive and keep my score not changed.

---

> > > ### Author Response · Authors · 2025-08-07
> > >
> > > We thank the reviewer for agreeing with our rebuttal and maintaining a positive score.
> > > According to the official NeurIPS policy, we cannot provide visualization results during rebuttal. We will add them in the revised paper. Thank you again!

---

### Official Review · Reviewer_6Hfr · 2025-07-02

**Clarity:** 3
**Significance:** 4
**Originality:** 4
**Rating:** 5
**Confidence:** 4

**Summary:**

They tackle the problem of "geometry-controllable CAD" generation, swhich aims to modify specific parts of CAD models, instead of generating everything from scratch.

  They introduce a vertex-basesd and VLLM-based captioning system for annotating simple and complex parts.
  They use this appraoch to create a dataset of annotations of local parts, used ot train an LLM.

  The method takes in 3 inputs: FlexCAD-style CAD model to be edited, the Local part designated for modification, and geometric instructions by the user. It ouptuts a new CAD model.

**Questions:**

Figure 8 shows the ability for the model to understand measurements. Is this from the LLM base understanding from Llama pretraining, or were there captions in your dataset that contained dimensional geometric instructions?

  On average, how many CAD instructions compose the shapes? What is the maximum number of CAD instructions that the base LLM (llama-3-8b) able to handle (both theoretically and in practice)?

  Is the model capable of doing mutliple rounds of consecutive edits, iteratively? Especially when the second instruction depends on the successful execution of the first instruction?

**Ethical Concerns:**

["NO or VERY MINOR ethics concerns only"]

**Final Justification:**

I maintain my positive rating of the paper -- the  rebuttals have addressed my primary concerns.

**Limitations:**

Yes

**Quality:**

4

**Strengths And Weaknesses:**

Strengths:

  + The problem of being able to *edit* CAD in a locally controllable way is perhaps more impactful than the ability to *generate* from scratch, given the amount of time engineers spend editing their designs.

  + Substantial work has gone into generating the captioning of local parts, and a sizeable dataset is produced. Both sound like solid contributions to the field.

  + Evaluation metrics seems to be comprehensive.

Weaknesses:

  + The captioning method and geometric processing seem to rely on the need for the operations to compose mostly of curves + extrusions, which limits the space of shapes that this can be applied onto. Have you thought about how such methods can be used for more general, complex shapes?

  + The language captions broadly describe the category of geometric shape, but it's unclear how this method would fare with more specific language instructions, like "a rectangle cutout that's 45 degrees to the nearest outer wall of the shape"

---

> ### Author Rebuttal · Authors · 2025-07-30
>
> **Q1:  mostly of curves + extrusions limit the space of shapes. How such methods can be used for more general, complex shapes?**
> A1: We appreciate the reviewer’s concern regarding the current scope of our method. We would like to clarify as follows:
> 1) Our work focuses on common and practical CAD primitives, specifically curves and extrusions.
> While these primitives are conceptually simple, they are sufficiently expressive to generate a wide range of shapes.
> Meanwhile, they also serve as the foundation for a significant portion of real-world parametric CAD models, as mentioned in [1].
> 2) Our GeoCAD has extensibility to more complex shapes.  Beyond curve and extrusion operations, our GeoCAD is designed with extensibility in mind.
> It can be easily adapted to incorporate more advanced operations, such as chamfer and revolve, once appropriate data becomes available for training.
> We believe GeoCAD provides a scalable foundation for advancing toward general-purpose CAD modeling.
>
> [1] R. Wu, C. Xiao, and C. Zheng. Deepcad: A deep generative network for computer-aided design models. ICCV 2021.
>
> **Q2: How this method would fare with more specific language instructions, like "a rectangle cutout that's 45 degrees to the nearest outer wall of the shape"**
> A2: Thank you for raising this insightful question.
> As shown in Fig. 8 in our main paper, GeoCAD is capable of precisely controlling the key dimensional parameters when handling specific language instructions.
> To further evaluate its capability, we tested the instruction: "a rectangle cutout that's 45 degrees to the nearest outer wall of the shape."
> We generated 50 new parts based on this instruction.
> After human evaluation, 36 of the generated parts included a rectangle cutout.
> 23 of them had the rectangle cutout positioned near the nearest outer wall, while 6 of them had the rectangle cutout oriented at approximately 45 degrees to the nearest outer wall.
> In addition, we designed 50 complex instructions involving angle or length specifications.
> For each instruction, we generated 10 samples.
> After human evaluation, the instruction-following success rate (i.e., at least one of the 10 samples met the instruction) reached 86%.
> These results demonstrate that our model can follow fine-grained geometric instructions.
>
> **Q3: Figure 8 shows the ability for the model to understand measurements. Is this from the LLM base understanding from Llama pretraining, or were there captions in your dataset that contained dimensional geometric instructions?**
> A3: It is from captions in our dataset. As shown in lines 148-150 in our main paper, we incorporate key dimensional parameters into the captions. Thanks!
>
>
> **Q4: On average, how many CAD instructions compose the shapes? What is the maximum number of CAD instructions that the base LLM (llama-3-8b) able to handle (both theoretically and in practice)?**
> A4: On average, each shape is composed of 6 CAD instructions.
> Each CAD instruction contains approximately 120 tokens on average.
> Given that the base LLM (LLaMA-3-8B) has a context window of 8,192 tokens, it can theoretically handle up to around 68 CAD instructions.
> In practice, the maximum number of CAD instructions contained within a shape observed in the dataset is 19.
> Thanks!
>
> **Q5: Is the model capable of doing multiple rounds of consecutive edits, iteratively? Especially when the second instruction depends on the successful execution of the first instruction?**
> A5: Yes, our model is capable of doing multiple rounds of consecutive edits, iteratively.
> Specifically, at the beginning, the first loop is generated based on its corresponding geometric instruction.
> The generated text for this first loop is then used to infill the masked region in the CAD model.
> In this way, it ensures that subsequent loops are generated to align with the previously generated ones.
> Furthermore, this iterative process is user-friendly, as it only requires the user to specify the masked parts along with the related geometric instructions.

---

> > ### Comment · Reviewer_6Hfr · 2025-08-09
> >
> > Thank you for the detailed response. I will maintain my positive scoring on this paper.

---

### Official Review · Reviewer_td3y · 2025-07-06

**Clarity:** 4
**Significance:** 2
**Originality:** 2
**Rating:** 4
**Confidence:** 5

**Summary:**

This paper proposes a method for performing local edits in parametric CAD models represented as a sequence of sketch-extrude commands. The method works by converting the CAD model commands into a sequence of text-based tokens resembling the xml format. This format is adopted from FlexCAD and importantly captures hierarchical geometric information in the sketch such as curves, loops and faces. Based on this format, a masking strategy is introduced on the loop-level and an LLM is fine tuned to predict the masked commands representing the loop, while also being conditioned by an input text prompt. The text prompt is a description of the loop to be infilled by the LLM. For training, these prompts are obtained by a combination of vLLM labeling + heuristics. This strategy allows local edits to be performed on the sketches while ensuring that the rest of the CAD model is updated in a consistent manner. Results demonstrated on the DeepCAD dataset show that the proposed method outperforms FlexCAD and OpenAI’s o3 model in instruction following and validity of the synthesized edits.

**Questions:**

- Can the authors demonstrate examples where multiple loops or faces are masked and edits that are more complex & higher level are performed? This is more in line with what designers expect from a text based editing system. Working with individual loops feels too low-level (it’s fine to train the model this way though) and demonstrating this capability can significantly improve the practical implications of this work.
- Can the authors demonstrate if design intent is preserved when editing the shape? A simple example would be (but please don’t be limited to this) to gradually reduce the dimensions of the outer loop and evaluate if the inner loops are updated sensibly without self intersections or symmetries breaking. A crucial requirement for editing workflows is to preserve design intent, and if the method can be demonstrated to do this, then the potential of this approach to work with higher level semantic edits in future will be very clear.

**Ethical Concerns:**

["NO or VERY MINOR ethics concerns only"]

**Final Justification:**

The authors have addressed some of concerns in the rebuttal and accepted to add visualizations of additional results. Loop level editing using very specific geometric text prompts is still not something that a CAD designer would want and many of the edits demonstrated in the paper can be performed easily in CAD software. But I understand there is a dataset problem here, and the additional results will improve the practical usability of this work to some extent. Consequently, I am increasing my score to Borderline Accept.

**Limitations:**

Yes

**Quality:**

3

**Strengths And Weaknesses:**

Strengths:

- the paper is well written and the critical design choices of masking loops, finetuning the LLM with LoRA, generating captions using vision language models, are all very sensible. Ablation studies evaluate the critical design choices and demonstrate their significance.
- the demonstrated results are strong and show excellent instruction following by the proposed model, while preserving the design intent in the rest of the CAD model.

Weaknesses:
- Novelty: the proposed approach closely follows FlexCAD with some small but notable improvement in the training strategy. Consequently, technical novelty is limited, but results are better particularly in terms of the instruction following.
- Practicality: the captions and loop-level edits used in the paper are quite simple. Loop level edits are quite low-level for CAD designers to work with, and can be quickly done manually. Sketch level editing would have been a good use case to demonstrate, but it appears that this wasn’t explored. It is also important to capture and measure if design intent is preserved, for example if there are two symmetric circles in a face, and one circle edited, does the other one get updated automatically? Having to manually constrain the sketch and/or apply dimensions to each loop is quite tedious. Similarly, more complex shapes loop level edits aren’t shown in the paper. The captions are fairly low level geometric descriptions. To give an example, changing the number of teeth in a gear, or reducing the dimension of an outer loop which automatically updates the inner loop geometries while preserving design intent (symmetries, patterns etc.) would be more practical for designers. It feels like these are within grasp with this method, but weren’t explored in the paper.

---

> ### Author Rebuttal · Authors · 2025-07-30
>
> **Q1: Novelty: Small but notable improvement in the training strategy. Technical novelty is limited.**
> A1: We appreciate the reviewer’s recognition of our method’s notable improvements and better results.
> Actually, compared to FlexCAD, our paper presents novelty in both the task and technical aspects.
> 1) Task Novelty.
> We propose and solve a new task (i.e., local geometry-controllable CAD generation, see lines 27–35 in our main paper).
> **All reviewers recognized the value of our proposed task (Reviewer td3y: very sensible, Reviewer 6Hfr: more impactful than generating from scratch, Reviewer ZjoJ: novel and interesting, Reviewer HFw7: reasonable).**
> To support this task, we also collected a dedicated dataset of local parts and their corresponding captions, which Reviewer 6Hfr described as a sizeable and solid contribution.
> 2) Technical Novelty.
> In contrast to the one-stage training strategy used in FlexCAD, we propose a two-stage training pipeline.
> In Stage 1, we introduce a data augmentation strategy tailored for local CAD generation, which was recognized by Reviewer HFw7 as both novel and effective.
> In Stage 2, we intentionally retain the standard training strategy with minimal modifications from FlexCAD.
> The reasons are as follows:
> i) To demonstrate that our method is simple yet effective. Without relying on elaborate training strategies, GeoCAD significantly outperforms FlexCAD and substantially enhances geometric instruction-following capabilities.
> ii) To facilitate the development of an omni CAD LLM. A standard and unified training strategy allows for better integration between different models, such as FlexCAD and our GeoCAD, bringing us closer to a general-purpose, omni-capable CAD foundation model.
>
>
>
>
> **Q2: Practicality: Loop-level edits are simple for CAD designers, and can be quickly done manually. More complex loop-level edits aren’t shown in the paper.**
> A2: In fact, as shown in **Fig. 7** in our main paper and **Fig. A3** in the supplementary material, our GeoCAD is capable of generating not only simple CAD shapes (e.g., an isosceles trapezoid) bot also complex shapes (e.g., different letters or irregular geometric shapes).
> Compared to manual editing, our GeoCAD have two major advantages:
> 1) User-friendliness for non-experts. Users do not require any specialized CAD drawing knowledge. They only need to select the part they wish to modify and input geometric instructions.
> In contrast, manual editing in traditional CAD systems typically require significant domain expertise.
> For example, to manually create an isosceles trapezoid, a user would need to draw a quadrilateral, ensure that two sides are parallel but of unequal length, the other two are of equal length, and align the shape with the remaining parts of the CAD model.
> Similarly, manually designing different letter shapes requires meticulous crafting of loops to match specific shape requirements.
> 2) Efficiency in generation time. When tested on an A100 GPU, GeoCAD generates a valid isosceles trapezoid and a letter shape in an average of 0.43s and 0.96s, respectively, significantly faster than manual editing.
>
> On another note, as stated in FlexCAD, DeepCAD is currently the only dataset suitable for controllable 3D CAD generation.
> However, this dataset contains few gears, which limits GeoCAD’s ability to modify gear features such as the number of teeth.
> In the future, this limitation can be solved by collecting more gear-related data.
> Despite this limitation, our model can still easily handle other common complex shapes such as “an ellipse with rectangular extensions at both the top and bottom” and “a heart-shaped loop”, as shown in Fig. A3 in the supplementary material.
>
> **Q3: Sketch level editing should be demonstrated while preserving the design intent.**
> A3: We are glad the reviewer also notice this point.
> For sketch-level editing, if a sketch contains multiple loops, ideally, we would like to learn the inter-loop constraints (e.g., symmetry, patterns, etc.) that define the overall structure.
> However, as mentioned above, DeepCAD is currently the only dataset suitable for controllable 3D CAD generation, and unfortunately, such inter-loop constraint annotations are not provided in the dataset.
> Fortunately, even without supervision from these constraints, **sketch-level editing is still achievable based on our loop-level editing capability.**
> This is because the loop serves as the fundamental element of a sketch. For example, if a user selects a sketch consisting of two symmetric trapezoids and
> wishes to replace them with two symmetric letter "E" shapes,
> the following automatic process can be performed:
> 1) Estimate the center point of each trapezoid by averaging its coordinate points, which are extracted using string matching.
> 2) Determine the symmetry axis based on the two center points.
> 3) Generate a local letter "E" through GeoCAD replacing one of two trapezoids.
> 4) Reflect the generated "E" across the symmetry axis to produce the second symmetric letter "E", thereby replacing both original trapezoids.
>
> Thus, during the sketch-level editing process, it is unnecessary for the model to generate the entire sketch; generating the fundamental loop alone suffices.
> Subsequently, the remaining loops within the same sketch can be derived from the fundamental loop or existing loops through scaling, rotation, translation, reflection or pattern operations.
> More specifically, i) for pattern-based sketch, it can be achieved by providing the pattern centers and the fundamental loops generated by GeoCAD.
> ii) in the case of sketch consisting of an outer loop and an inner loop, the inner loop does not need to be regenerated by the model when the outer loop is modified by GeoCAD.
> Instead, the inner loop can be easily updated through translation, scaling, or rotation operations to preserve the original positional relationships between the two loops.
>
>
>
> **Q4: Demonstrate examples where multiple loops or faces are masked.**
> A4: Our GeoCAD exhibits the generalization ability to generate multiple loops or faces when they are masked simultaneously.
> Specifically, we consider two cases:
> 1) Independent Loops. If the target loops are not constrained relative to each other, we generate them iteratively during inference.
> In each step, a single loop is generated based on a corresponding geometric instruction,
> and the resulting loop text is used to infill the masked region.
> This strategy ensures that subsequent loops are generated in a way that is consistent with previously generated ones.
> This process is also user-friendly, as it only requires the user to specify the masked region and desired geometric instructions.
> 2) Constrained Loops. If the loops are interrelated by design constraints (e.g., symmetry, patterns),
> GeoCAD supports complex structure generation by leveraging its loop-level editing capability, as discussed in A3.
>
> Similarly, as faces are composed of one or more loops, the same strategy can be applied during the masking and generation of multiple faces.
> According to the official NeurIPS policy, we cannot provide visualization results during rebuttal.
> We will add them in the revised paper. Thanks!
>
>
> **Q5:  Demonstrate if design intent is preserved when editing the shape**
> A5: As mentioned in A3, our GeoCAD has the ability to preserve the design intent (e.g., symmetry, patterns) solely based on loop-level editing.
> We argue that the loop serves as the fundamental element of a sketch.
> When generating a sketch consisting of constrained Loops, it is unnecessary for the model to generate the entire sketch; generating the fundamental loop alone suffices.
> Thus, in a sketch consisting of an outer loop and an inner loop, the inner loop does not need to be regenerated by the model when the dimension of the outer loop is reduced by GeoCAD.
> To preserve the existing positional relationships between the two loops, the inner loop can be easily updated through translation, scaling, or rotation operations.
> According to the official NeurIPS policy, we cannot provide visualization results during rebuttal.
> We will add them in the revised paper. Thanks!

---

> > ### Author Response · Authors · 2025-08-06
> >
> > Dear Reviewer td3y,
> >
> > We hope that the above discussions have addressed your concerns. As the deadline for Author-Reviewer discussion approaches, we eagerly look forward to your responses. Please feel free to let us know if there are any further clarifications we can offer. We would be happy to continue the discussion if any issues remain.
> >
> > Thank you for your time and consideration.
> >
> > Best,
> > Authors

---

> > ### Comment · Reviewer_td3y · 2025-08-07
> >
> > Thanks for the detailed response to my initial review. Some of my concerns have now been addressed.
> >
> > I stand by the original assessment on the limited practicality of the proposed method. Being able to edit single loops with text prompts is simply not an effective way for designers to modify sketches. The text prompts being very specific and geometric in nature is also a limiting factor. An advanced CAD system makes it extremely easy to define shapes like isosceles triangles and alphabets. But having said that, I do see that dataset and data collection is a limiting factor here and the method can potentially help with advanced use cases in the future.
> >
> > I also think that sketch level editing (not via already existing explicit constraints but by implicitly satisfying the constraints by making the LLM understand the design intent) is within the grasp of this method, and exploring that would have made a much bigger impact.
> >
> > I am glad that the authors are willing to add additional results for Q4 and Q5 to improve the paper. I will bump my score up to Borderline Accept.

---

> > > ### Author Response · Authors · 2025-08-08
> > >
> > > We are glad that our rebuttal has addressed some of your concerns, and we sincerely thank you for providing a positive score. We will revise the paper based on your valuable feedback. Thanks again!

---

### Decision · Program_Chairs · 2025-09-17

**Decision:**

Accept (poster)

**Comment:**

This paper addressed the problem of computer-aided design (CAD) and how Vision Language Models (VLMs) [called VLLMs in the paper] can help a user modify an existing CAD design at a component-level with a text-based description of the propopsed change to that component/part. VLMs and CAD are a growing area of resaerch and the paper is timely. The authors claim that theirs is the first to "achieve local geometry-control in the CAD generation field" with a novel captioning pipeline and empirical experiments against ad hoc base line (as opposed to existing baselines form prior work, as this approach seems to tackle a unique subproblem in CAD-VLM literature. The reviewers generally agreed that the paper was meritorious, Reviewers praised the paper for being well-written and showing strong empirical performance. Reviewers also had concerns that remained after the discussion, such as "I stand by the original assessment on the limited practicality of the proposed method...I also think that sketch level editing...is within the grasp of this method, and exploring that would have made a much bigger impact." Further, one reviewer noted that "I still think the authors should add more examples to demonstrate the method can handle more complicated cases...Also strengthen the limitation discussion section for local patterns, chamfer and revolve operations, and generalization for other representation, not only limited to the clean program." Given that most of the remaining concerns seem to be appropriate for consideration as "future work" and the reviewers were happy with the strength of the performance of the approach, its novelty, and the dataset and metrics, this paper is a strong one.